# Growing in Scarcity: Pre-Hispanic Rain-Fed Agriculture in the Semi-Arid and Frost-Prone Andean Altiplano (Bolivia)

Pablo Cruz [1,*], Richard Joffre [2,*], Thibault Saintenoy [3] and Jean-Joinville Vacher [4]

1 UE CISOR CONICET, Universidad Nacional de Jujuy, San Salvador de Jujuy 4600, Argentina
2 CEFE, University of Montpellier, CNRS, EPHE, IRD, 34090 Montpellier, France
3 Instituto de Ciencias del Patrimonio (Incipit), CSIC, 15707 Santiago de Compostela, Spain; thibault.saintenoy@incipit.csic.es
4 Institute of Research for Development, UMR 208 Paloc, 13572 Paris, France; jean.vacher@ird.fr
* Correspondence: pablocruz@conicet.gov.ar (P.C.); richard.joffre@cefe.cnrs.fr (R.J.)

**Abstract:** Ancient Andean agricultural landscapes have been the subject of a large number of archaeological and agro-ecological studies, which generally refer to regions with favourable environmental conditions or, in the case of arid and semi-arid environments, those with irrigation facilities. The aim of this article is to present and analyse the pre-Hispanic rain-fed farming systems widely represented in two adjacent regions of Bolivia's arid and cold southern Altiplano. The search for archaeological agricultural areas combined aerial analysis and field surveys. Agro-ecological characterisation was based on historical and ethnographic studies of the region's present-day populations. Despite their geographical proximity, similar environmental conditions, and same agropastoral way of life, the typology of cultivated areas developed in the southern altiplano differs significantly. Within this same framework of adaptation and resilience, the sectorisation of agricultural systems observed in these two regions reveals a regional productive specialisation that favoured internal exchanges and exchanges with other regions. These differences are related to two models of non-centralised, low-inequality societies—one strongly based on cohesion and the other characterised by greater fragmentation and social conflict—underlining the limits of strict environmental determinism in shaping agricultural landscapes. These results provide new food for thought in the debate on the use and value of rain-fed agricultural practices and more broadly on the diversity of adaptations by human societies in extreme and unstable environmental contexts.

**Keywords:** Andes; landscape mapping; pre-hispanic agriculture; rain-fed cultivation; traditional ecological knowledge

## 1. Introduction

The archetypes of high-altitude agriculture, such as the ancient agricultural landscapes of the Andes, have been the subject of a large number of archaeological and agro-ecological studies. Taken together, these studies reveal a wide variety of agricultural landscape types, including the imposing terraced slopes found in the valleys of Peru, the raised fields systems around Lake Titicaca, or the dense network of cultivation enclosures at Coctaca in the Quebrada de Humahuaca in Argentina, among many other examples [1–5]. Most studies of ancient Andean agriculture focus on regions with environmental conditions favourable to the growth of rain-fed crops or, in the case of arid and semi-arid environments, those with sufficient water resources for irrigation (see, for example, [6–8]). Areas characterised by low annual rainfall (<450 mm) are generally considered unsuitable for current cropping practices when there is no irrigation [9]. However, both in the southern Bolivian altiplano, where annual rainfall is below this threshold, and in many arid regions of the world, rain-fed crops are present [9–11] and contradict this preconception.

Furthermore, unlike other arid and semi-arid regions of the world, the vast majority of which are located in hot climates, farming societies in the Andean highlands have had to

develop strategies to cope with a combination of two major risks: drought linked to low and irregular rainfall, and frost [12–15]. Due to its low latitude and high altitude, the risk of frost is high throughout the year [16]. Several strategies have been implemented to manage the risks associated with cold weather and, more generally, climatic risks [17]. On the one hand, the plots are located and laid out in such a way as to take advantage of the drainage of cold air along the slopes and to reduce the risk of frost [18]. On the other hand, a set of practices are linked to cultivation (e.g., choice of sowing dates and biennial fallowing) and the use of a wide agrobiodiversity of varieties or cultivars with distinct agroclimatic requirements [17,19–21]. In the Intersalar region of the southern Bolivian altiplano, we have highlighted the existence of a pre-Hispanic agricultural system operating entirely on a rain-fed basis [22]. Surprisingly, this large-scale agricultural development, centred on quinoa cultivation and capable of generating productive surpluses, enabled a local society to prosper during a period when environmental conditions, already extreme, were further exacerbated by the confluence of different climatic events on a global scale.

This is not a unique case. Indeed, our research in the nearby Carangas region, also characterised by an extreme high-altitude climate, reveals a significant development of pre-Hispanic agriculture, albeit with notable differences from the Intersalar region. Despite the absence of irrigation and the apparent rusticity of farming practices, agriculture was maintained in these agricultural zones over a long period of time. All these practices are part of a traditional ecological knowledge (TEK) adapted to these particular conditions. The implementation of a set of adapted and robust farming techniques, based on detailed knowledge of environmental conditions, testifies to the adaptation and resilience of local populations to the region's harsh environmental conditions. Such adaptations of societies in regions subject to severe or highly variable environmental conditions have often been described for small-scale societies [23] or pastoral societies [24], but much more rarely for societies practicing rain-fed agriculture [9,25–29].

The aim of this article is to present and analyse the ancient rain-fed agricultural systems that predominate in a vast area of the southern Bolivian Altiplano, not yet described, in an attempt to understand how they were able to overcome the difficult conditions of the region. The study of the agricultural landscapes of Carangas represents a new milestone in our understanding of the various solutions devised by the populations over the centuries to practice perennial rain-fed farming in ecological conditions that exclude them a priori. These results broaden our vision of traditional Andean agriculture, mainly characterised by regions benefiting from a favourable environment or water resources for irrigation. More broadly, all the cases dealt with shed new light on the debate on the adaptations of agricultural societies in extreme and unstable environmental contexts.

## 2. Materials and Methods

### 2.1. Study Area and Climate

The study area concerns the part of the central altiplano located between the Sajama Volcano to the north, the Salar de Uyuni to the south, the western Cordillera to the west, and the Poopó Lake and the town of Pampa Aullagas to the east (Figure 1). It includes the regions known as Carangas and Intersalar of the departments of Oruro and Potosí in Bolivia. This area is articulated by vast plains between 3850 m to the north and 3650 m to the south, with hills dominating the surface of these plains by around 300 m, and mountainous formations and high peaks, of which the Sajama volcano is the highest point (6542 m).

Located between latitudes 18° and 20° south and at a high altitude, the region studied is characterised by a cold, arid tropical climate. The precipitation gradient varies from 350–450 mm to 100–200 mm yr-1 from north to south of the region, following a marked north–south gradient [30] (Figure 2). The rainfall distribution is unimodal during the austral summer months, but there is a differentiation between the north and south of our study area in terms of the duration of the rainy season. In the north of the zone, it lasts 100–110 days between mid-November and mid-March, while in the south, the rainy season is shorter (80–90 days), centred on the months of December to February [14]. The average



annual temperature is 7.4 °C, with a marked effect of altitude on relief. On the plains outside the mountains, a small east–west temperature gradient corresponds to the steady rise in altitude from Lake Poopó to the Cordillera Occidental (Figure 2b). Whatever the season, the daily temperature range is greater than the seasonal range and can reach up to 25 °C, leading to frost risks throughout the year. In October and November, frosts have a major impact on crop seedling mortality, and the east–west gradient already identified for average temperatures is marked (Figure 2c). In April, below freezing temperatures can have a major impact on crops at harvest time, with the same east–west gradient (Figure 2d). Climate fluctuations have been documented throughout the late Holocene, showing changes on centennial and multi-decadal scales, but also strong interannual variations driven by the El Niño Southern Oscillation [31].

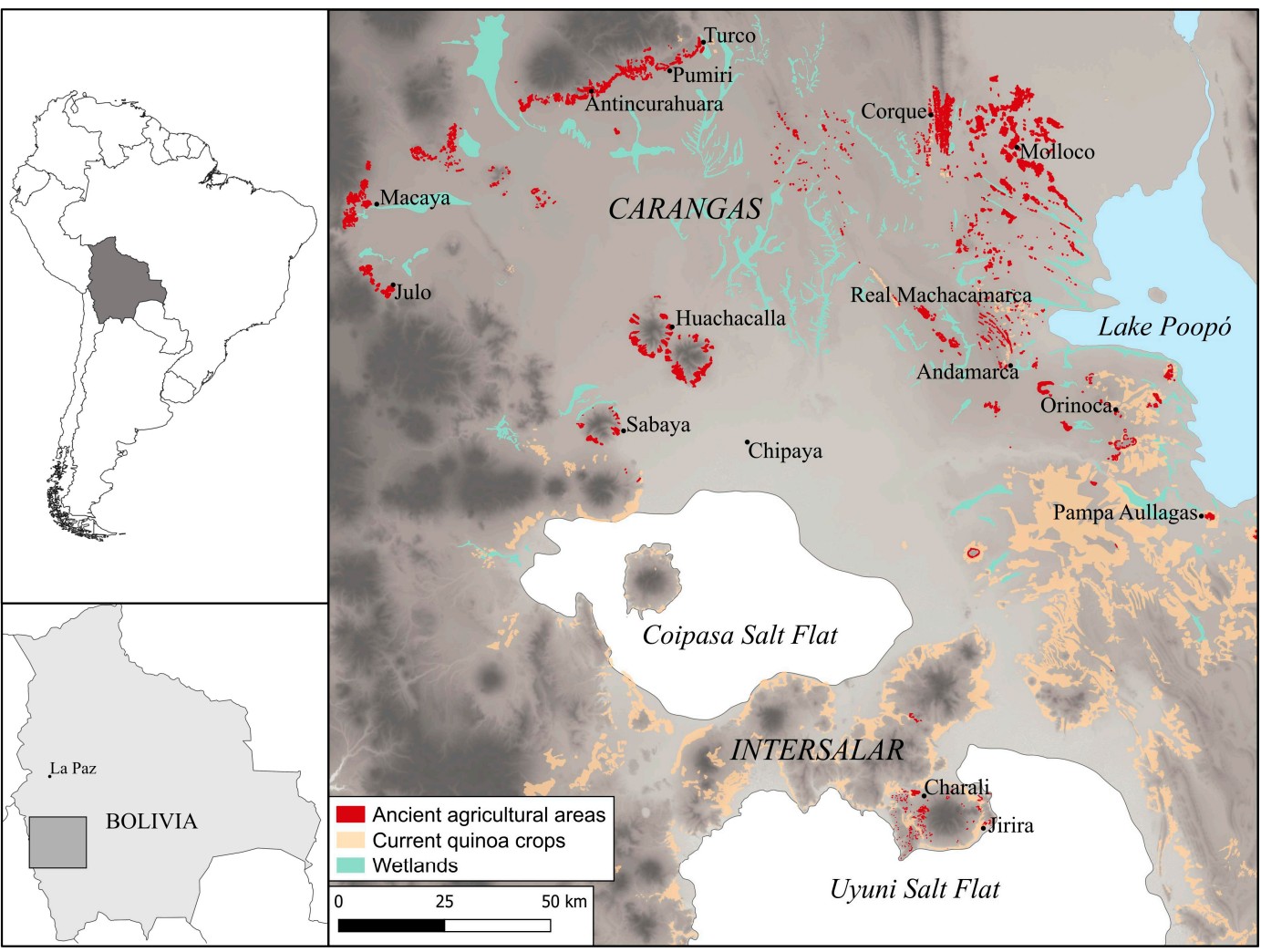

**Figure 1.** Locations of Carangas and Intersalar regions, Bolivia.

The region's current population is predominantly indigenous, mainly Aymara-speaking and, to a lesser extent, of the Uro-Chipaya language family, whose way of life is traditionally agropastoral. At present, the economy of the Intersalar region is based on the traditional cultivation of quinoa, the region being the world's leading producer. On the other hand, llama and alpaca farming predominate in the Carangas region, as do, to a lesser extent, potato and quinoa cultivation.

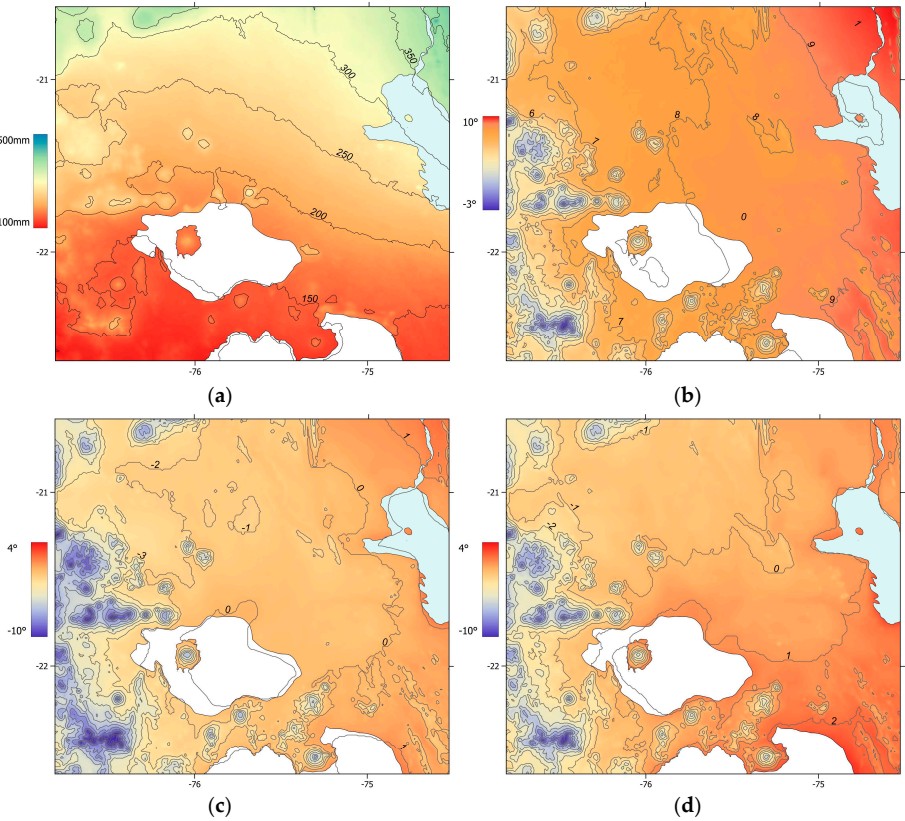

**Figure 2.** Spatial distribution of monthly and annual climate parameters obtained from WorldClim version 2 at 30 s spatial resolutions and for 1970–2000 [30]. (**a**) Mean annual precipitation, (**b**) mean annual temperature, (**c**) October average daily minimum temperatures, and (**d**) April average daily minimum temperatures.

### 2.2. Archaeological Background

Archaeological studies indicate in this part of the altiplano the presence of populations that adopted an agropastoral lifestyle as early as the formative period (1500 BC). From the Late Regional Development Period (LRDP, 1200–1450 CE) onwards, the population of the region grew considerably, with the emergence of regional societies with very well-defined characteristics that were maintained, with varying degrees of change, throughout the subsequent late and colonial periods (15th–19th centuries). During the course of the LRDP, notable differences were noted between the Intersalar and Carangas regions, in terms of settlement patterns, funerary models, and ceramic styles. These differences are consistent with the information provided by colonial sources, which record the settlement of different ethnic groups in this part of the altiplano. In Carangas, the ethnic group from which the region takes its name, and in the Intersalar, the Aullagas-Huruquillas populations became part of the Quillacas-Asanaque ethnic federation [32]. The study of several Intersalar dwelling sites from this period has shown that the local populations had a non-centralised form of social organisation with relatively low levels of inequality, resulting from mechanisms of cooperation and social cohesion [33]. In the mid-fifteenth century, the region was colonised by the Inkas and incorporated into Qullasuyu, the southern quadrant of the Empire. In Carangas, the Inkas had a significant presence, with important settlements such as Antincurahuara, ceremonial centers such as Waskiri [34] and Changamoco, and groups of polychrome tombs. During this period, the Intersalar region underwent a significant depopulation, probably linked to the transfer of a large part of the local population to other regions to be used as labour in mining and agricultural centers [35]. At the end of the 16th century, the local populations of both regions were affected by the introduction of the mita system, the compulsory taxation of labour in the mines of Potosí [32,36].

### 2.3. Agricultural Background

Despite severe climatic constraints (see above), since pre-Hispanic times, the Carangas and Intersalar regions are major producers of Andean tubers and grains, mainly potato and quinoa. The presence of productive agriculture in these regions since pre-Hispanic times is based on TEK, in particular very detailed local topoclimatic knowledge and simple soil management techniques [22], as well as on the use of the great biological diversity of potato and quinoa varieties [37,38]. Specific strategies have been devised by local people to deal with the two major constraints: the risk of frost and insufficient rainfall. Annual rainfall in the Carangas and Intersalar regions does not cover the water requirements of a complete cycle of quinoa and potato crops. The concentration of water over time was achieved via the practice of biennial fallowing allowing the accumulation of water resources over 2 years [22]. This practice requires the soil texture to be sandy or sandy-silty. Biennial fallow is now practiced in all quinoa crops in this region of the southern altiplano. Detailed knowledge of the spatial heterogeneity of the risks of frost by populations is the second fundamental aspect of agricultural management in the regions under consideration [22]. The current management of agricultural land is based on each farmer having access inside the community territory to a wide range of contrasting ecological situations and, more generally, on land management practices that minimise frost risks.

Potato farming is characterised by the use of several *Solanum* species, the most widely cultivated of which, *Solanum tuberosum* subsp. *andigenum*, *S. juzepczukii*, and *S. curtilobum*, show a wide range of adaptations to agroclimatic constraints [38]. Due to their glycoalkaloid content, *S. juzepczukii* and *S. curtilobum* are known as bitter potatoes. *S. juzepczukii* is remarkably resistant to low temperatures, down to $-5$ to $-7\ ^\circ$C, compared with only $-2\ ^\circ$C for *S. tuberosum*. In the Carangas region, known for its cold nights, *S. juzepczukii*'s luki variety has been present as basic food since pre-Inka times [39]. The numerous tubers of *S. juzepczukii* are generally small in size and contain high levels of glycoalkaloids, making them unsuitable for direct consumption. They are processed in a traditional and unique process of freeze–drying by successive freezing and dehydration, made possible by the alternation of very cold nights and very sunny days during the Altiplano winter. It removes the glycoalkaloids from bitter potatoes and produces processed, edible tubers known as *chuño*. This makes *chuño* a long-lasting (several decades) and easily transportable form of storage (three-quarters lighter). This food reserve is strategic in a region where the probability of crop failure is high. This process has been used in the Andes for over 2000 years [40]. In his chronicles of 1553, Cieza de Leon [41] emphasised the main role played by the *chuño* in the diet of the peasants of the Altiplano and its importance in the food trade: "The *chuño*, highly prized and of great value.... many Spaniards became rich and then left for Spain very prosperous with the sole commercial activity of bringing the chuño to Potosi" [41] (p. 124).

### 2.4. Methods

Between 2007 and 2018, an interdisciplinary research programme on pre-Hispanic agricultural systems combining archaeology and agro-ecology was carried out in the Intersalar region over 1800 km$^2$ [22]. Between 2018 and 2023, we extended this programme to the ancient agricultural systems of the neighbouring Carangas region, covering a total of 3000 km$^2$ (Figure 3). The first step was a systematic preliminary analysis of high-resolution satellite imagery (GeoEye, DigitalGlobe, CNES/Astrium, and CNES/Airbus) over the study area to identify possible areas of ancient cultivation. The spatial patterns observed enabled us to delineate three sub-regions within the Carangas region as a whole (see the Results section below). We then built a QGIS geographic database using land surface topography extracted from the Shuttle Radar Topography Mission (SRTM) v2 digital elevation model (DEM) with a resolution of 1 arc second (~30 m) downloaded from the Reverb | ECHO website. Data from climate maps (WorldClim version 2 [30]), topographic maps, and place names were also used and integrated into the GIS database. The areas of archaeological

cultivation were digitised by photo-interpretation of satellite images using QGIS editing tools (http://www.qgis.org/ (accessed on 2 May 2024)).

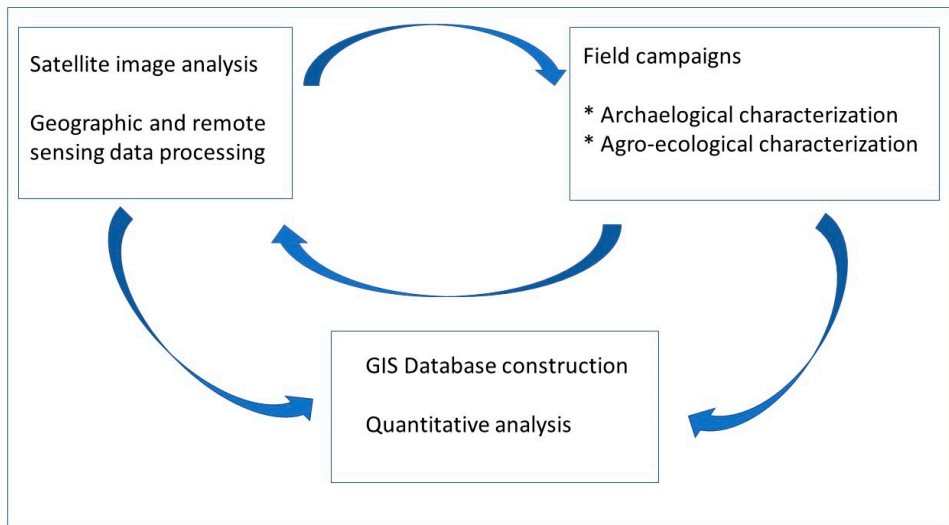

**Figure 3.** Workflow chart.

Five successive systematic field surveys covering all the Carangas sub-regions were then planned and carried out over the five-year period. The campaigns were organised to verify, correct, and complete the preliminary maps drawn up from the analysis of satellite imagery and to describe and characterise the identified crop areas in detail. These detailed surveys were carried out using unmanned aerial vehicles to establish precise photogrammetric and detailed topographies. At the same time, agro-ecological characterisations, archaeological studies, and material analyses (ceramic fragments and lithic agricultural implements—shovels and picks) were carried out in the various areas selected. After each field campaign, we integrated our field data into the QGIS database. This process was carried out iteratively after each field campaign in order to correct the database and prepare for the next campaign. These tasks enabled us to identify and measure all the ancient culture and allows for quantitative analysis (Figure 3). These field studies were supplemented by an exhaustive review of documentary sources and historiography relating to the region, and ethnographic interviews were conducted with the current inhabitants [42]. Although the agricultural areas studied date back to pre-Hispanic times, a small number of them are still farmed today. Interviews with local producers were essential for understanding how the different farming areas identified functioned.

## 3. Results

### 3.1. Ancient Pre-Hispanic Agricultural Systems of the Arid Altiplano

In both the Carangas and Intersalar regions, the ancient cultivated areas are located mainly on the slopes and foothills of the mountains, with virtually no ancient agricultural areas on the flatlands of the Altiplano. The absence of ancient crops on the plains is linked to the mayor frost risk due to the flow of cold air coming down from the slopes (katabatic drainage), as shown for the entire Bolivian Altiplano [16] and in the Intersalar region [18]. However, most of the new cultivated zones accounting for the recent exponential growth of quinoa cultivation are localised in the plains. Behind this evolution are two factors: warming temperatures minimising the frost risk and easier use of agricultural machinery in these flat areas [21].

The recurrent presence on the surface of pre-Hispanic ceramic remains and lithic shovels, indicate that the agricultural surfaces date in all cases from the period of LRDP and the Late Period, between the mid-13th century CE and the mid-16th century CE. In the Carangas region, one of the most explicit indicators of the pre-Hispanic chronology of agricultural areas lies in the presence of ceremonial pukaras, characterised by a variable

number of concentric walls located on low hills but visible from afar. The material evidence of ritual and ceremonial practices is concentrated in the space of these sites, where the figures of the wak'a were probably preserved, tutelary entities that governed agricultural production and the fate of the people [43]. A fundamental aspect to consider here is that, although the formation of these productive spaces dates back to pre-Hispanic times, numerous documentary sources refer to a continuity of agricultural practices in both regions during the later Colonial and Republican periods [32,36]. In fact, current crops are observed in all ancient agricultural areas, although they only represent between 1% and 5% of the total surface areas.

Both in Carangas and in the Intersalar region, almost all of the ancient cultivation areas were rain-fed. The few permanent streams available in this part of the Altiplano are used to develop and maintain wetlands, the grazing and watering of which are vital for camelid farming. This practice continues to this day, particularly in Carangas, where camelid farming remains very important to the regional economy. A recent paleoenvironmental study of a sediment sample from a wetland in the Intersalar region determined that this practice of wetland creation and maintenance dates back to the LRDP [44] and is consistent with the archaeological record of the region [22].

Although rain-fed agriculture in these regions is almost exclusively concerned with the production of potatoes and quinoa, it is interesting to note the very marginal presence of maize cultivation. This is surprising, given that this crop is considered typical of warmer environments with greater water availability. Maize cobs have been found in collective tombs dating from 977 +/− 30 years CE to 1178 +/− 68 CE [45] and local varieties of maize continue to be grown on a very small scale in different localities in Carangas and Quillacas [46,47] (Figure 4).

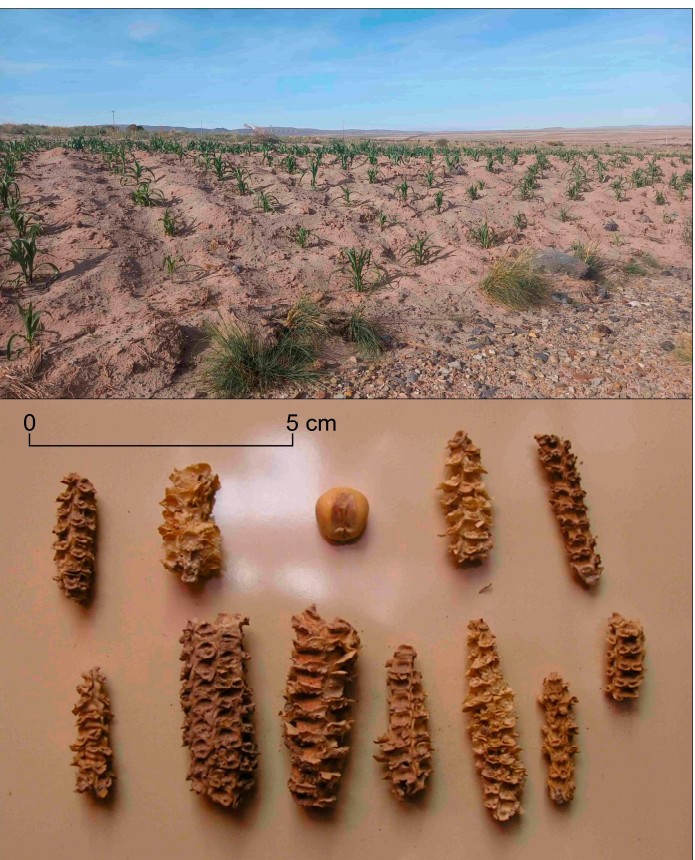

**Figure 4.** (**Above**) Current rain-fed maize cultivation in Quillacas (photo: Alejandro Bonifacio). (**Below**) Altiplanic maize stalks found in a funerary context dated 1100 CE, Escaramayu, Salar de Uyuni, Bolivia [45].

*3.2. The Pre-Hispanic Agricultural Surfaces of Carangas*

The agricultural areas of Carangas analysed here cover 30,300 ha, mainly cultivation enclosures (canchones) (95%), i.e., plots bounded by stone walls around 1,5 m high. These plots are commonly identified by the local inhabitants as *inka uyo*, in Aymara: fence or corral of the Inka, in allusion to their pre-Hispanic age. To a lesser extent, there are various types of terraced land, collectively known as *inka taqana*. Just over half of these old plots (52%) are located on gentle slopes (less than 10%) and in the foothills, with the remainder on steeper slopes and on the tops of low hills. Most of these plots have an orthogonal contour, usually rectangular. Their surface area varies considerably, from a few dozen to 3000 m$^2$, with an average of 620 m$^2$. Only a few sectors have irregular plots, which are generally smaller in size.

Over and above their formal similarities, these ancient agricultural areas have characteristics that enable us to distinguish three clearly differentiated types of agricultural landscape: the first type is located to the west of the central Carangas high plateau, the second to the east of this plateau, and the third in the plains to the north-west of Lake Poopó. We are not considering here the areas to the north of the Coipasa salt flat, where environmental conditions have severely limited the development of agriculture.

3.2.1. Western Sector of Carangas

The agricultural areas between Turco, Pumiri, Macaya, and Julo are divided into two distinct sectors in terms of the types of areas cultivated. The larger area is mainly located on the lower slopes of the Turco mountain range and the Asu Asuni, Chullkani, Chilliri, and Putintika hills. In this sector, the predominant agricultural layout is made up of a series of rectangular plots arranged longitudinally in the direction of the slopes, which vary between 10% and 34% (Figure 5). The width of the plots is generally between a quarter and just over half their length. They are bounded by stone walls around their entire perimeter. In most cases, the lower wall is also used as a retaining wall, which can be more than a meter high. The plots located in the Turco mountain range (Pumiri and Antincurahuara) are characterised by a large number of *despiedres*, or stone clearing heaps of small and medium-sized stones resulting from intensive stone removal.

In these plots, the slope is not uniform between the top and bottom, with the steeper upper part serving as an impluvium for the very flat lower part, where the concentration of rainfall due to the lower retaining wall will provide better moisture conditions and deeper soils for crops. The sectorial distribution of humidity allows crops to develop despite the water constraints. The shape and layout of these plots contrasts with the classic model of agricultural terraces, designed to create totally horizontal cultivation areas. Within these groups of plots, there may also be groups of longitudinal plots laid out in the direction of the slope but not terraced. In general, the width of these narrow plots is between 4 and 15% of their length, which can be several hundred meters. It is relevant to highlight here that these areas of cultivation with plots on hillsides are concentrated in a sector of Carangas where the Inka presence was particularly evident. Several of the features observed in the area (distribution of plots, impluvium, stone clearing heaps, and longitudinal fields) were also identified in a large agricultural settlement established by the Inkas in the Quebrada de Humahuaca in north-western Argentina [5]. This does not mean that the Carangas agricultural zones were created by the Inka state, but rather that they were an optimisation of the pre-existing local agricultural system.

The second sector is located between the towns of Macaya and Julo, where there are several agricultural areas that differ significantly from the previous systems. These areas are mainly located on the slopes of small hills, alluvial fans, and plateau with a gentle gradient (5–7%). They consist of groups of rectangular and irregularly shaped plots, often slightly terraced, with small longitudinal plots and large unconditioned areas.

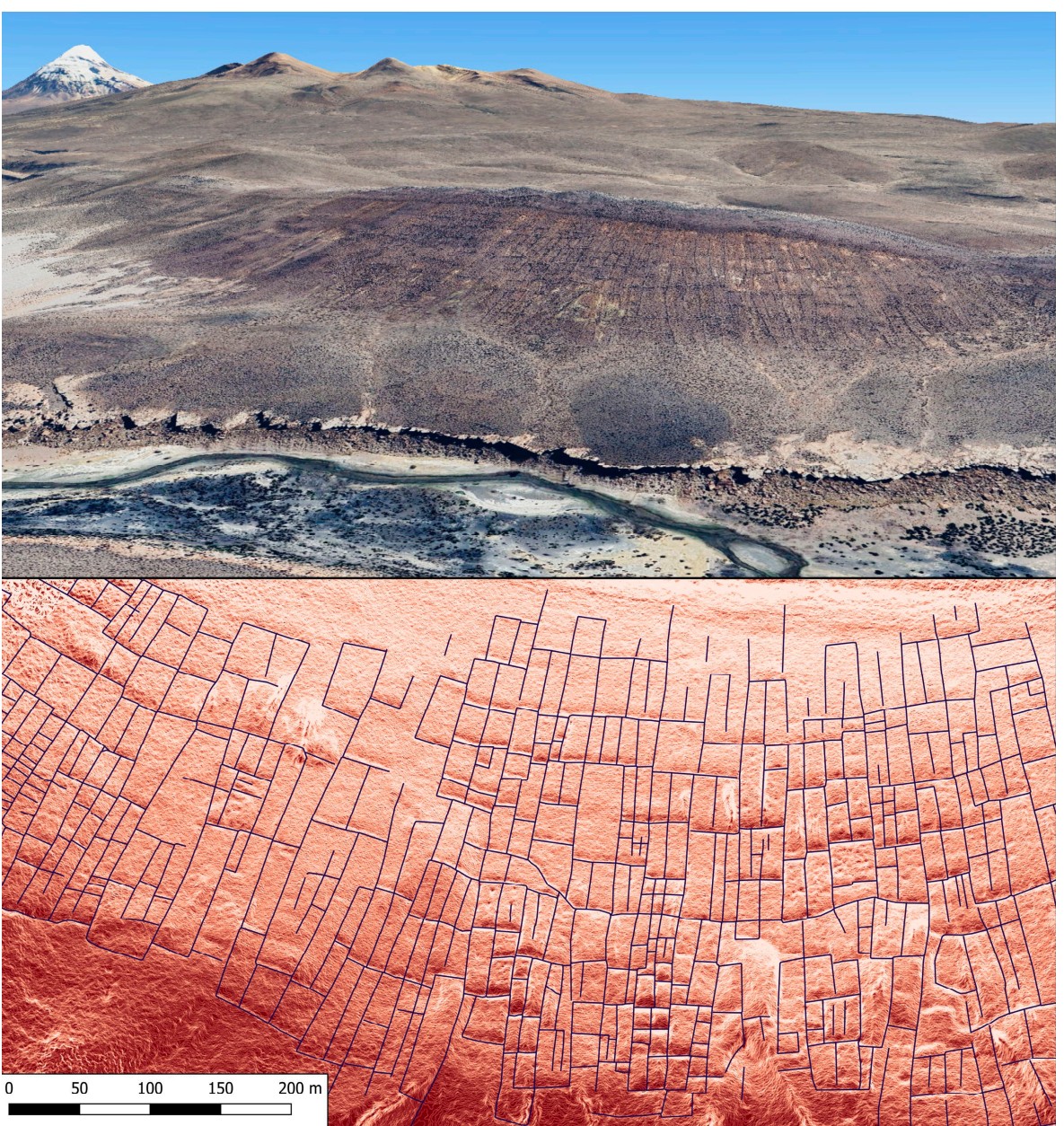

**Figure 5.** Cultivation enclosures in the vicinity of the Sajama volcano. (**Above**) Three-dimensional view of agricultural ancient plots with the Sajama volcano at the left side (source: Google Earth). (**Below**) Map of the plots delimited by stone walls.

### 3.2.2. Eastern and Central Sector of Carangas

This area includes the Corque, Andamarca, Orinoca, and Pampa Aullagas mountain ranges to the east and the Huachacalla mountain range in the centre. The area is characterised by intensive use of cultivable landforms, particularly those with the greatest capacity to capture and store rainwater and provide protection against frosts. As a result, we find the greatest diversity of pre-Hispanic cultivation surfaces—regular plots, the most common; irregular plots; and, to a lesser extent, different types of terraced surfaces: transverse canals, linear and contour terraces, and terraced plots. Within this large zone, however, there are differences, with certain types of parcels dominating. Thus, while the Andamarca, Orinoca, and Huachacalla mountain ranges are dominated by canchones on regular slopes, irregular canchones dominate in the Pampa Aullagas region (Figure 6), and contour terraces and terraced canchones are in the majority in Real Machacamarca (Figure 7).

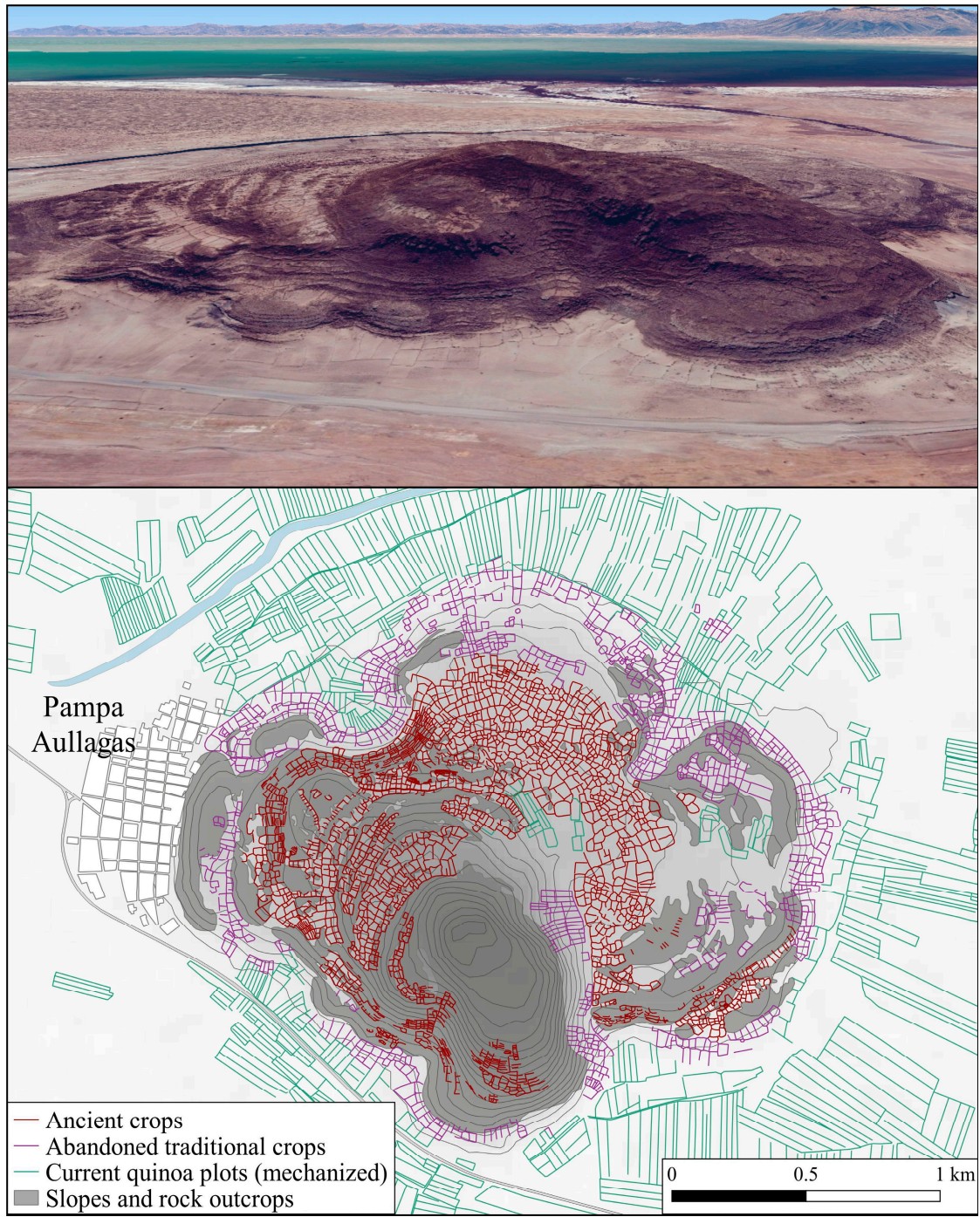

**Figure 6.** Pampa Aullagas agriculture plots. (**Above**) Three-dimensional view of the agricultural area with Lake Poopó in the background (Source Google Earth). (**Below**) Map of ancient and current crop plots.

The Corque chain of mountains is a remarkable example of intensive land use. This is an agricultural area located at the top of a mountain range crossed longitudinally by numerous parallel rock outcrops. These formations have been intensively exploited to create spaces for cultivation in the interstices, resulting in the formation of a dense conglomerate of plots covering some 1620 ha. Limited by the distance between rock outcrops, these plots are significantly smaller than the plots mentioned above, ranging in size from 30 m$^2$ to 250 m$^2$ and averaging 103 m$^2$ (Figure 8).

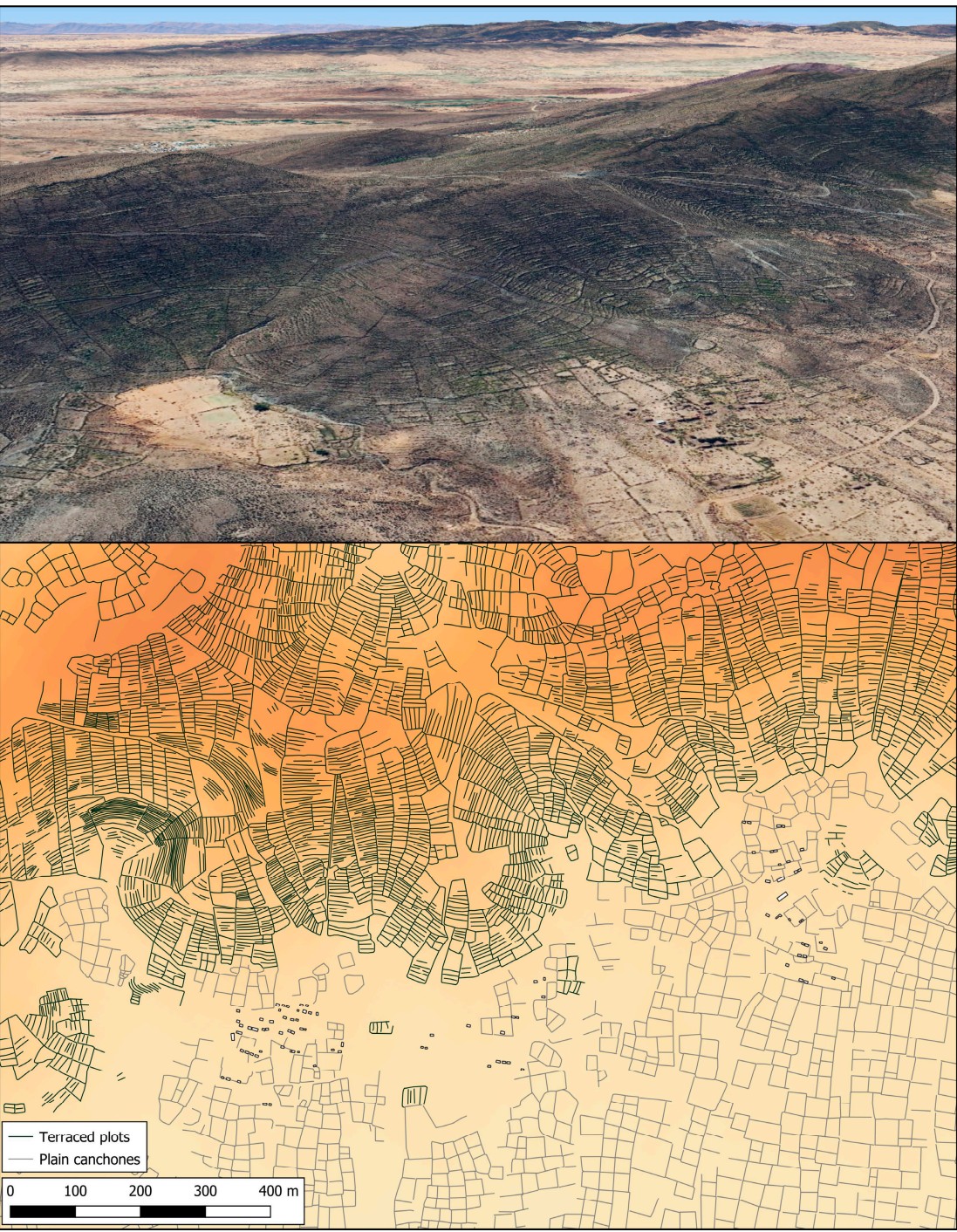

**Figure 7.** Real Machacamarca agriculture terraces and plots. (**Above**) Three-dimensional view of the sector (Source Google Earth). (**Below**) Map of the terraces and plots.

In all the agricultural areas of the sector, the foot of the slopes and surrounding plains are generally occupied by cultivation plots, mostly of orthogonal contour. These plots constitute the most frequent type of cultivated area in the Carangas region, covering more than 5000 ha; are also the ones that mostly continued to function until recent times; and, to a much lesser extent, are still cultivated today. In this sense, although the frequent presence of lithic shovels and ceramic remains testifies to the agricultural use of these sites in pre-Hispanic times, the plots that can be observed today are the result of a constant dynamic of creation and readjustment of the cultivation spaces.

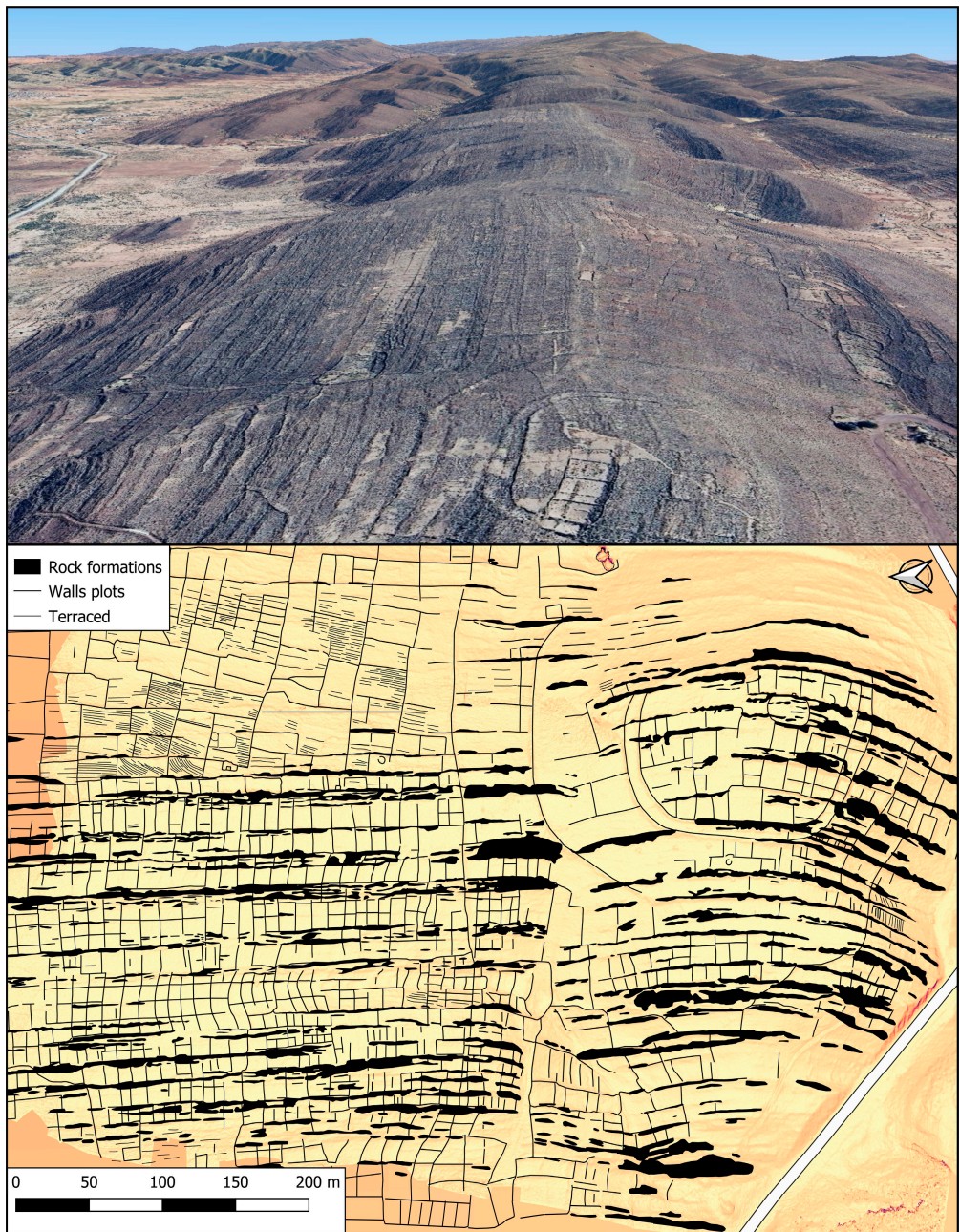

**Figure 8.** Ancient agriculture plots near Corque. (**Above**) Three-dimensional view of the ancient agricultural plots (Source Google Earth). (**Below**) Map of plots delimited by rocky outcrops and stone walls.

### 3.2.3. Northwest Plains of Lake Poopó

Unlike the two previous sectors, this area is limited to the plains to the north-west of Lake Poopó, where the presence of a water table less than 5 m deep is attested to by the many wells observed, and where the risk of frost is less than in the west of our study region. The particular cultivation system found in this area is called tajllitas. Tajllitas are quadrangular or rectangular-shaped cultivation corrals bounded by high walls (Figure 9), generally built with champa, the top layer of soil, and ichu (Stipa ichu), a grass typical of the Altiplano. Once they had been built, the tajllitas were used as corrals to shelter the animals, with the high earthen walls releasing the heat accumulated during the day at night, thereby reducing the risk of radiative frost. Later, when a sufficient layer of guano accumulates over several months, the plots are used as cropping areas and the animals are

moved to other, uncultivated plots. The size of the plots is linked to the number of animals, and the length of the cultivation cycle is generally of four years. Each plot is rotated after four years. According to current agricultural knowledge, the most common rotation is as follows: two years of potato cultivation, one year of quinoa cultivation, and used as corral for the animals in the fourth year. The organic fertilisation provided by the animals helps to restore the fertility of the soil after the three years of cultivation and before the next cycle.

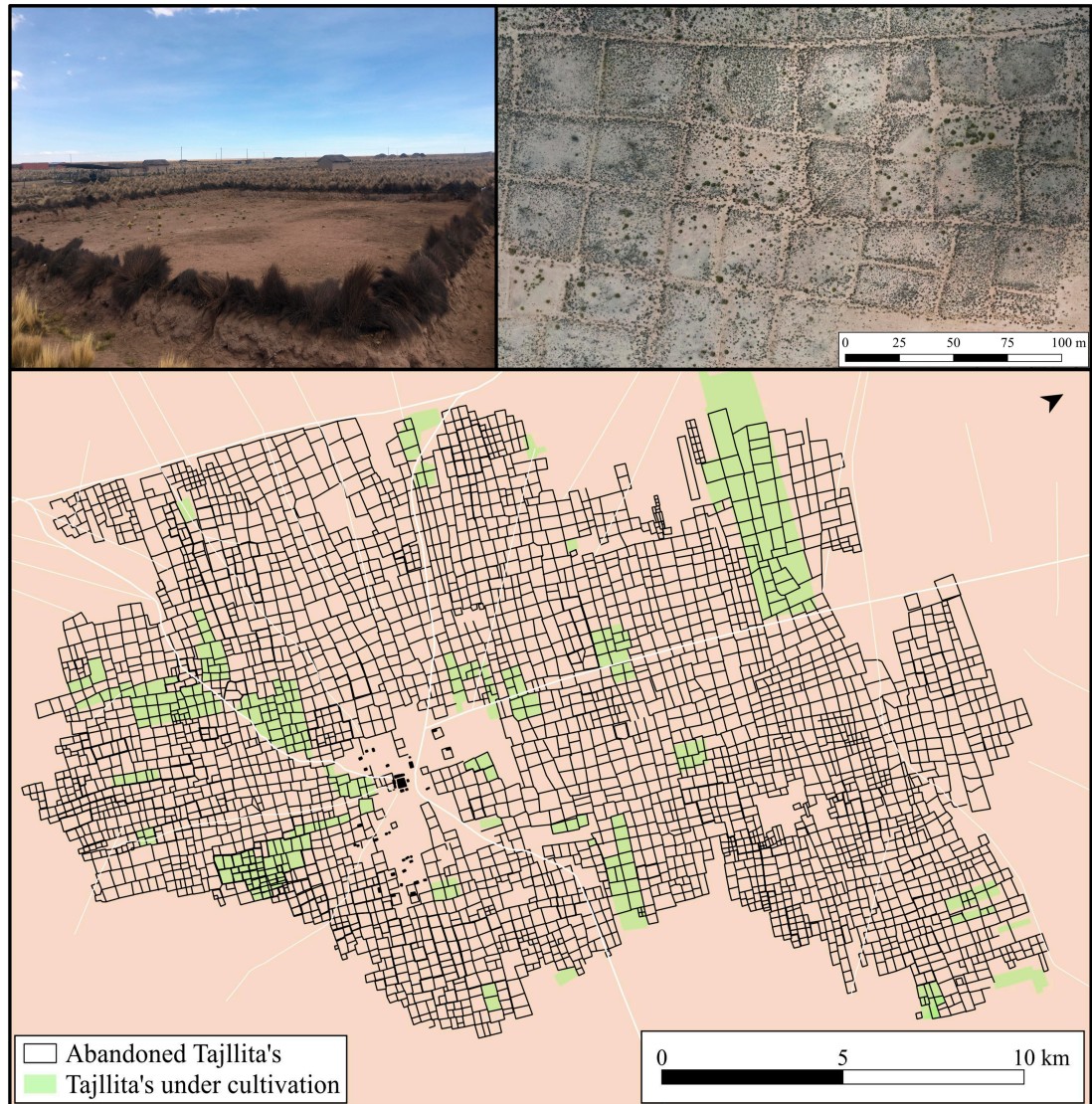

**Figure 9.** Map of Molloco's tajllitas agricultural system. (**Above**) Photograph and aerial view of tajllita plots (photo: Jean Vacher and Pablo Cruz). (**Below**) Map of Molloco tajllitas sector.

The field records carried out in the localities of Molloco allowed us to determine that the antiquity of this particular modality of cultivation in corrals dates back to the LRDP. However, as in the cases discussed above, this system remained in force over time, processes that led to the formation of vast reticular conglomerates that can exceed 500 ha, occupying a total area of more than 10,200 ha.

### 3.3. The Pre-Hispanic Agricultural Surfaces of the Intersalar Region

Situated between the Uyuni and Coipasa salt flats at an altitude of over 3700 m, the Intersalar region combines vast plains and mountainous terrain. Archaeological research carried out since 2007 has revealed the development of a pre-Hispanic society during the 13th and 15th centuries, a period when environmental conditions, already extreme in

this region, deteriorated in terms of drought and cold [22]. In a study area of 1700 km², 49 habitation sites were identified and attributed to this period. One aspect that stands out from these sites is the high number of granaries, with a minimum of 7607 granaries recorded across the 49 sites. Excavations and test pits at several of these sites indicate that these structures were used to store quinoa grains. Today, this Intersalar region is one of the world's main quinoa production and export centers [48].

The populations that inhabited the Intersalar region during this period developed a model of rain-fed agricultural production based on a modification of the landscape that was markedly different from that of the neighbouring Carangas and Quillacas regions. In fact, the ancient agricultural production areas are mainly made up of micro-terraces that form the small productive spaces, between 1 and 10 m², which we call minimal surfaces of production (MiSP) [22]. MiSPs densely occupy all the favourable spaces on the slopes and—like any other earthwork structure—make them suitable for cultivation by slowing down surface runoff, thereby promoting soil and water retention. Their construction did not require much labour, sometimes consisting of a few stones in a row. Linear terraces, requiring a greater investment in labour, were also observed, but much more rarely. In areas with a gentle slope (typically <5°), these conditioned spaces are combined with sloping terraces and vast fields, which are found either in groups or in isolation (Figure 10).

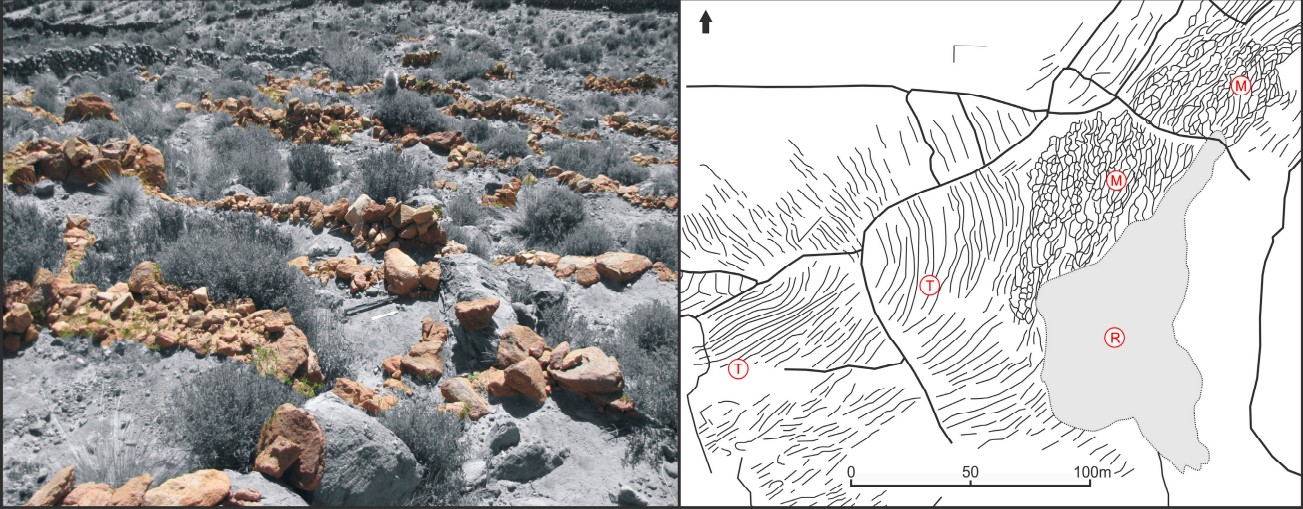

**Figure 10.** Ancient agricultural areas close to the Charali archaeological site in the Intersalar region. (**Left**) Minimal surfaces of production areas (MiSPs) with highlighted stone alignments (photo Richard Joffre). (**Right**) Landscape organisation showing archaeological lineaments (M, MiSP; R, rock outcrop; T, sloping terraces). Source: modified from [22].

Field surveys and a thorough analysis of satellite images made it possible to identify 2670 ha of ancient small cultivation areas (MiSPs), most of which are located between 3700 m and 4000 m, on slopes between 2° and 30°, and on surfaces mostly oriented between NW and SSW. Modelling the MiSP distribution using identified areas through the Random Forest methodology with environmental and topoclimatic variables and distance to the dwelling sites as predictors lead to over ≈17,000 ha of MiSP over the Intersalar region [49]. In addition to these minimal areas of production, the plains adjacent to the slopes are home to crop patches similar to those observed in Carangas and Quillacas, although in smaller quantities (620 ha). This modelling of productive areas is consistent with the number of granaries identified in the same area, which are used to store quinoa grains. It should be noted that a very small proportion of these agricultural areas have been reused for potato cultivation in more recent times. According to the region's oral memory, this is evidenced by the presence of pirwas, potato storage structures, located inside the ancient cultivation areas. The decline of traditional markets and the new quinoa boom led to a drastic reduction in potato cultivation in the mid-1980s.

## 4. Discussion

### 4.1. Adaptation, Resilience, and Regional Complementarity

While all human societies have been able to generate responses to climatic constraints which, with varying degrees of impact, are present in all regions of the planet, the scope for response is narrower in regions with arid and cold environments, such as the South-Andean Altiplano, and these responses have a decisive impact on lifestyles. In fact, agricultural production in these regions depends on obtaining sufficient water resources to develop a complete crop cycle during the favourable period, i.e., excluding periods of severe night frost. As annual rainfall does not cover the water requirements of a complete crop cycle, efficient farming in these regions is only possible if the necessary quantity of water can be obtained by concentrating water resources either spatially or temporally. In terms of spatial concentration, irrigation increases the availability of water for crops over small areas by means of impluvium, spring catchments, and canals. No major canal-type infrastructure exists in the Intersalar and Carangas regions to supply crop-growing areas. Spring catchments are marginally present in the regions studied and only to supply very small areas of irrigated gardens of a few dozen square meters, located around the villages. The few permanent streams available in this part of the Altiplano are used to develop and maintain wetlands, the grazing and watering of which are vital for camelid farming. Finally, in the Turco region, we described areas of differentiated impluvium within the plots (Figure 4). This feature, very rare in the Andes, had already been encountered in a large agricultural settlement established by the Inkas in the Quebrada de Humahuaca [50]. Solutions based on the spatial concentration of water resources, generally considered to be the most frequent and interesting in all arid regions of the world, are therefore largely marginal here. The practice of biennial fallowing used here enables water resources to be accumulated over two years. This means that there is a significant stock of water in the soil at the time of sowing, irrespective of the year's rainfall. It has been shown for wheat that the amount of water in the soil at the time of sowing is closely linked to yield. This practice is not specific to the Andean region and is also found in various dry agricultural regions of the world [10,22,51]. It is part of a set of practices linked to the crop itself, in particular the choice of species and varieties, taking advantage of the huge biodiversity of Andean cultivars against drought and frost risk. In addition, and in response to the inter-annual variability of the start of the rainy season, plant density and sowing dates are flexible within certain limits so that the full cycle can take place and so that there is sufficient water for germination and to fill the grains or tubers. With regard to optimal water management, the traditional Andean practice of sowing in clumps offers better growing and harvesting conditions, as has been demonstrated in other arid regions [52,53].

The agricultural developments presented here highlight the particular way in which local populations have adapted and succeeded in overcoming difficult environmental conditions. These adaptations call into question the direct interference of climate on social processes, an environmental determinism that has been widely questioned [54–56]. In fact, these productive developments pose a number of paradoxes. Firstly, despite the aridity of the climate and the frequency of decadal droughts, the region's agricultural systems were, and continue to be today, mainly rain-fed. The few permanent streams available are used to improve grazing areas, the pastures of which are used to rear llamas and alpacas. This allows us to understand the implications of the agropastoral way of life adopted by local populations since pre-Hispanic times, and the vital importance of camelids for subsistence, the economy, and regional interactions. Secondly, the agricultural developments discussed here were shaped during the LRDP (12th–15th centuries), together with a population increase and the flourishing of well-defined societies in both regions, a time when environmental conditions, already extreme, worsened considerably in terms of drought and cold [22].

Despite their proximity and similar environmental conditions and agropastoral way of life, the rain-fed agricultural systems developed in both regions differ markedly. This has to do with two fundamental and interlinked principles that have always governed

agricultural practice in this part of the Andes: climate risk mitigation, and economic and social complementarity. In this extreme environment, slight variations in rainfall patterns and in the intensity and frequency of night frosts can have enormous consequences, leading to crop losses. The diversification of cropping areas and the maintenance of a high level of crop genetic diversity [26] reduce the risk of climate variability affecting all crops. Significantly, the development of rain-fed agriculture in the region has gone hand in hand with other adaptation and resilience mechanisms that have enabled the region to cope with, and even benefit from, difficult environmental conditions [42].

Firstly, numerous landraces of potato; quinoa; and other crop species more suited to other ecological environments, such as maize, which are not only adapted to rain-fed cultivation in a highly water-scarce environment but also resistant to frost and low temperatures that prevail for much of the year, were created [38,57]. In the particular case of the quinoa varieties that continue to be rain-fed in the region, it is important to note their variability in flowering and ripening time within the same population, as well as their significant ginning after panicle ripening [58]. These characteristics, which in other production contexts could be detrimental, reduce the risk linked to climatic agents. On the other hand, the use of cold and dry air in the processing of potatoes, a staple food in these regions, to guarantee their storage over long periods, is a conservation technology specific to the Andean highlands that dates back to pre-Hispanic times [40,59]. This involves the use of low night-time temperatures during the dry austral winter (June and July) for the dehydration and freeze–drying process of bitter potatoes, resulting in two distinct products—*chuño* and *t'unta*—as well as the dehydration of camelid meat by exposure to dry air and strong solar radiation to produce *charqui* [60–62]. The fact that these two conservation practices persist in Altiplano societies is particularly interesting, as they take advantage of climatic features that guarantee food security, even though these periods could severely affect harvests and animal husbandry.

Within this same framework of adaptation and resilience, the sectorisation of agricultural systems observed in Carangas, more oriented towards potato cultivation, and in the Intersalar region, centred on quinoa cultivation, reveals a regional productive specialisation that, without excluding other crops for consumption, favoured internal exchanges and exchanges with other regions [63]. The ecological complementarity between this part of the altiplano and neighbouring regions with mesothermal valleys is remarkable. To the west, on the other side of the Cordillera Occidental, are the high Andean valleys and oases of northern Chile, and to the east, those of Norte Potosí and Yura. In fact, important agricultural centers are located on the eastern margin of Lake Popoo, where environmental conditions are more favourable and water resources allow crops to be irrigated. In this respect, historical documentation shows consolidated interactions between Carangas and northern Chile in the early colonial period [64], most likely in continuity with pre-Hispanic times [65,66]. Similarly, various sources from the early colonial period refer to the fact that settlers from the former ethnic and territorial jurisdiction of the Quillacas, who constituted the Intersalar, owned agricultural land in the warm valleys of Yamparaez in Chuquisaca [32]. In both cases, population and production archipelagos were established according to the Andean model of vertical control of ecological levels first proposed by John Murra [67].

### 4.2. Two Models of Resilient Societies

In a recent work, we related the adaptive capacities and resilience of the pre-Hispanic population that flourished in the Intersalar region during the 12th–15th centuries CE, with a sustainable model of non-centralised society, with low levels of inequality and where mechanisms of social cohesion prevailed [33].

Several indicators suggest that the pre-Hispanic inhabitants of the Carangas region had a similar model of a non-centralised society with relatively low levels of inequality, which is particularly evident mainly in the large number of house-tombs (chullpas), as well as in the known habitation sites [68,69]. Indeed, low inequality is also pointed out

by the Gini coefficient calculations based on the areas of 218 dwellings from three LRDP sites yielding an average of 0.232, which is very close to the average Gini of 0.234 obtained from 12 sites in the Intersalar region [33]. However, there were notable differences between the pre-Hispanic Carangas and their Intersalar neighbours. The pre-Hispanic populations of the Carangas settled mainly around villages with houses built of perishable materials (Figure 11), which suggests greater mobility of the populations, perhaps linked to the importance of camelid livestock. In contrast to the invisibility of the habitation sites, the aforementioned large number of tomb houses, mostly built with adobe, stand out in the Carangas landscape. Instead, while pre-Hispanic habitation sites in the Intersalar region are highly visible, the funerary record is characterised by burials located in rocky overhangs, often imperceptible. Likewise, in Carangas, we have recorded more than 130 ceremonial pukaras directly associated with ancient agricultural areas [44], a non-existent phenomenon in the Intersalar region. The fortified nature of ceremonial *pukaras* should be emphasised here, as they probably housed the tutelary *wak'as* on whom the fertility of crops and livestock depended [34]. As Bouysse-Cassagne and Chacama [65] point out, these wak'as were objects of covetousness and theft during ritual disputes. Even if the fortifications of these religious sites may have had a more symbolic and dissuasive function, which we cannot yet know, the high number of ceremonial *pukaras* recorded at Carangas indicate that the fragmentary nature of this society was accompanied by a high level of social conflict at the local level.

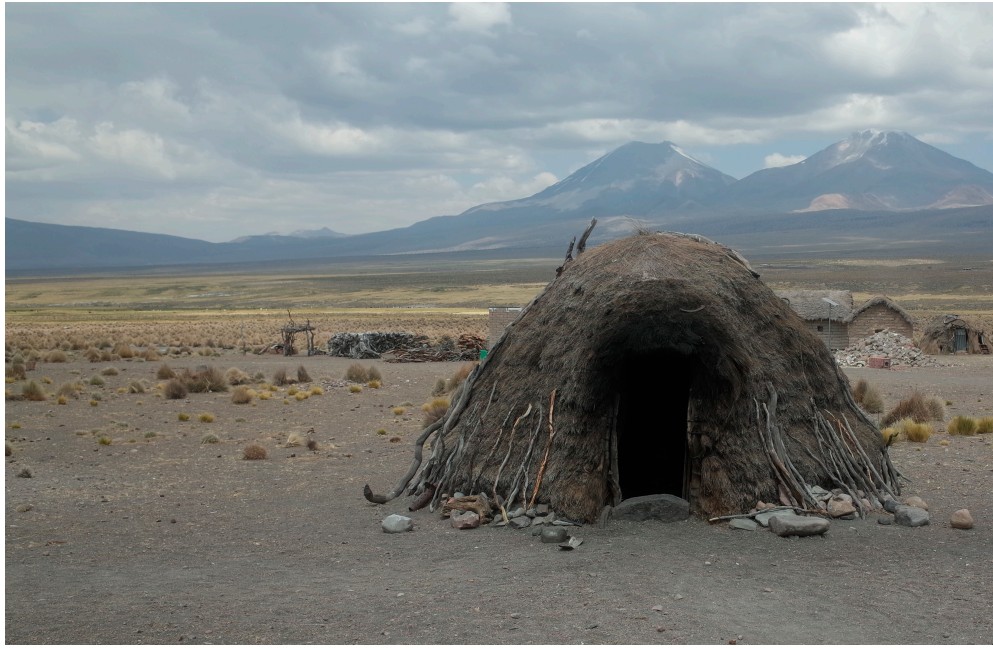

**Figure 11.** Photograph of one of the few traditional Carangas dwellings built from perishable materials but still standing today (Photo Pablo Cruz).

The differences between the Carangas and Intersalar regions are very significant insofar as they reveal, despite their geographic, environmental, and lifestyle proximity, two models of society—one strongly based on cohesion (Intersalar) and the other characterised by fragmentation and greater social tension (Carangas)—which is a clear example of the limitations of environmental determinism.

## 5. Conclusions

A fairly widespread conception of agriculture in arid and semi-arid environments makes it conditional on the availability of sufficient water resources for crop growth, whether supplied by rainfall or supplemented by irrigation canals. Consequently, the development of rain-fed agriculture is considered typical of environments that are naturally

favourable to crops. However, several studies have shown how societies in different arid regions of the world have been able to develop efficient rain-fed agricultural systems [9,25–29]. In line with these studies, our results highlight that pre-Hispanic rain-fed agriculture is widely present in a frost-prone and arid region of the southern Andean altiplano.

The almost absolute predominance of rain-fed agriculture throughout the region is based on precise, in-depth knowledge of the micro-environmental conditions; the biological diversity of Andean cultivars being taken into account; and a range of land management practices designed to minimise the environmental risks of soil erosion and climatic adversities. Taken as a whole, this knowledge is very rich and constitutes an adapted form of traditional ecological knowledge. Risk mitigation is thus achieved through the diversification of production and the economic and ecological complementarity between agriculture and camelid breeding. In summary, the diversity of pre-Hispanic agricultural developments in Carangas and the Intersalar region demonstrate the ability of local populations to overcome, and even take advantage of, difficult environmental conditions. The effectiveness of these productive developments is reflected in their permanence over a long period. And, although only a small part of these ancient cultivation areas is still cultivated today, the current inhabitants are heirs to a know-how accumulated through centuries of experimentation, a collective knowledge that has allowed, for example, the Intersalar region to become the main producer of traditional quinoa in the world.

**Author Contributions:** Each author contributed to the research and work presented in this article. The contributions of each include the following: Conceptualisation, P.C. and R.J.; Methodology, P.C. and R.J.; Formal analysis, P.C.; Investigation, P.C., R.J., J.-J.V. and T.S.; Writing—original draft preparation, P.C., J.-J.V. and R.J.; Writing—review and editing, P.C., R.J., J.-J.V. and T.S.; Visualisation, P.C. and R.J. All authors have read and agreed to the published version of the manuscript.

**Funding:** We thank the French National Research Institute for Sustainable Development (IRD) for logistical support and funding. This research was partially supported by funds from Axencia Galega de Innovación (GAIN, Ref. IN607D 2023/1).

**Data Availability Statement:** The data that support the findings of this study are available from P. Cruz and R. Joffre on request.

**Acknowledgments:** This research is the product of a joint project between the National Scientific and Technical Research Council (CONICET, Argentina), the National Centre for Scientific Research (CNRS, France), and the French National Research Institute for Sustainable Development (IRD, France). We thank the National Direction of Archaeology in the Bolivian Vice-Ministry of Culture for having authorised and supported this research, and the IRD Representation in Bolivia for their collaboration with the project.

**Conflicts of Interest:** The authors declare no conflicts of interest.

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
