# Peer review of "Growing in Scarcity: Pre-Hispanic Rain-Fed Agriculture in the Semi-Arid and Frost-Prone Andean Altiplano (Bolivia)"

_land, doi:10.3390/land13050619_

Round 1
Reviewer 1 Report
Comments and Suggestions for Authors
I have carefully reviewed the paper “Growing in Scarcity: Pre-Hispanic Rain-Fed Agriculture in the Semi-arid Andean Altiplano (Bolivia)”. The topic is interesting, but the work needs significant improvement. Notably, the novelty of this study is somehow unclear. The clarity and depth of the methodology section are also severely lacking.
The introduction should offer a more detailed explanation on study's originality, outlining precisely how it contributes to our understanding of the field.
The methodology section requires significant enhancement. I would suggest providing a comprehensive workflow, outlining the steps they took in conducting this research, including the specific methodologies employed. Without this information, it's challenging to assess the validity and reliability of the study.
The authors should offer more details about the data used in their analysis. Mere mentions of satellite image analysis and field surveys are inadequate for a scientific research paper. Please let your leaders know how these methods were applied and the specific data collected is essential.
The subsection titled "Ancient pre-Hispanic agricultural systems of the arid Altiplano" appears to be a summary of previous works rather than the authors' own findings. This section should be relocated to the study area section, as it doesn't contribute to the presentation of the study's results.
The authors need to clarify how they calculated the percentages and areas mentioned in lines 242, 245, and 246, providing more detail in the methodology section.
Reviewer 2 Report
Comments and Suggestions for Authors
Dear Authors
I enjoyed reading this manuscript. It is, apart from minor issues, well written and interesting.
Main issues to consider:
1. The distinction between results and discussion is not completely clear, as is often the case with archaeological writing. I acknowledge that it is not easy to clarify, even to yourselves, what is direct interpretation (part of the results) and what is the extended interpretation (discussion). You may wish to reconsider the location of some parts of the text (e.g., L. 199-202), or maybe simply use a single “Results and Discussion” section, and it with a subsection of “broader implications” (or another phrasing to that effect).
2. You may wish to consider – for this manuscript or for the future – comparing your findings to the desert agriculture in the Negev (Israel) and southern Jordan from the 3rd-9th centuries CE, which share some interesting similarities, but also with some striking dissimilarities.
3. The figures you provided are very informative. The manuscript can probably benefit from more of these, especially ones that indicate specific features that you mention in the text.
4. Section 4.2 (L. 510-543) feels too speculative. It seems that more work is needed to support the claims.
Minor comments:
L. 98: “Below-freezing” instead of “negative”
L. 105: Uro-Chpaya language family?
Fig. 2: Please provide a colour scale
L. 160: “Multi-species”, ‘diversity” and “several” all mean the same.
L. 165-166: Problematic grammar. And note how you spell “Inca” instead of “Inka” previously.
L. 217-224: Just for your knowledge: Bedouins in the Middle East also often sow wheat or barley in rain-fed plots, to secure forage for their livestock. So maintaining vegetation for forage in marginal regions is a phenomenon.
L. 329-336: So, there seems to be a “downhill” shift in agricultural land-use (see also Figure 5). Can you elaborate on this?
L. 339-355: Very clever. Worth further exploration.
Figure 8: There is no “C” in the legend and no “R” in the figure.
L. 424-426: Not that surprising – see my first comment.
Comments on the Quality of English LanguageEnglish is fine.
Round 2
Reviewer 1 Report
Comments and Suggestions for Authors
I would like to thank the authors for their effort in revising the paper. But, I must point out that a significant portion of my previous comments has not been fully addressed.
In the response to the reviewers, the authors claimed to have enhanced the introduction section pointing out the pioneering nature of the research, both thematically (Andean rainfed agricultural systems, a topic very little explored) and regionally (the ancient agriculture of the semiarid altiplano). However, upon review, it appears that only two sentences were added to the introduction. Furthermore, the second sentence lacks a citation and contains a grammatical error (unlike others should be unlike other).
Additionally, I have previously requested a workflow chart to better understand the methodology used in this study; however, this request has not yet been addressed by the authors.
There is also a need to correct the section numbering in the methodology part of the manuscript. The current numbering does not logically reflect the sequence of content; for instance, 'Study Area and Climate' should be numbered as 2.1 instead of 1.
Regarding my comments on “the calculation of percentages and areas mentioned in lines 242, 245, and 246”, the authors have stated that these areas were calculated though an "exhaustive analysis of satellite images." To ensure transparency and allow for a thorough evaluation by reviewers, it is crucial that the authors provide a detailed description of this analysis process in the methodology section. Providing such detailed methodological insights will significantly enhance the reader's and reviewers' understanding of the procedures and strengthen the study's validity.
Comments on the Quality of English Languageline 66: others should be other
